# The Effect of Health Worker Training on Early Initiation of Breastfeeding in South Sudan: A Hospital-based before and after Study

**DOI:** 10.3390/ijerph16203917

**Published:** 2019-10-15

**Authors:** Justin Bruno Tongun, James K Tumwine, Grace Ndeezi, Mohamedi Boy Sebit, David Mukunya, Jolly Nankunda, Thorkild Tylleskar

**Affiliations:** 1Centre for International Health, University of Bergen, 7804 Bergen, Norway; zebdaevid@gmail.com (D.M.); Thorkild.Tylleskar@uib.no (T.T.); 2Department of Paediatrics and Child Health, College of Medicine, University of Juba, P.O. Box 82 Juba, South Sudan; 3Department of Paediatrics and Child Health, School of Medicine, College of Health Sciences, Makerere University, P.O. Box 7072 Kampala, Uganda; kabaleimc@gmail.com (J.K.T.); gndeezi@gmail.com (G.N.); jnankunda@gmail.com (J.N.); 4Department of Internal Medicine, College of Medicine, University of Juba, P.O. Box 82 Juba, South Sudan; mbsebit@gmail.com

**Keywords:** Baby-Friendly Hospital Initiative, training, breastfeeding initiation, South Sudan

## Abstract

Globally, suboptimal breastfeeding contributes to more than 800,000 child deaths annually. In South Sudan, few women breastfeed early. We assessed the effect of a Baby-Friendly Hospital Initiative training on early initiation of breastfeeding at Juba Teaching Hospital in South Sudan. We carried out the training for health workers after a baseline survey. We recruited 806 mothers both before and four to six months after training. We used a modified Poisson model to assess the effect of training. The prevalence of early initiation of breastfeeding increased from 48% (388/806) before to 91% (732/806) after training. Similarly, early initiation of breastfeeding increased from 3% (3/97) before to 60% (12/20) after training among women who delivered by caesarean section. About 8% (67/806) of mothers discarded colostrum before compared to 3% (24/806) after training. Further, 17% (134/806) of mothers used pre-lacteal feeds before compared to only 2% (15/806) after training. Regardless of the mode of birth, the intervention was effective in increasing early initiation of breastfeeding [adjusted prevalence ratio (APR) 1.69, 95% confidence interval CI (1.57-1.82)]. These findings suggest an urgent need to roll out the training to other hospitals in South Sudan. This will result in improved breastfeeding practices, maternal, and infant health.

## 1. Introduction

Optimal breastfeeding includes early initiation of breastfeeding (within one hour of birth), exclusive breastfeeding (EBF) from birth up to six months and continued breastfeeding for at least two years [1]. According to the 2016 Lancet Breastfeeding Series, attaining high rates of EBF could prevent as much as 800,000 child deaths per year globally [2,3]. Results from the Neovita study in Ghana, India and Tanzania reported that early initiation of breastfeeding independently reduces neonatal mortality and facilitates EBF [4].

A meta-analysis of Demographic and Health Surveys (DHS) from 29 sub-Saharan African countries showed that 50% of women practiced early initiation of breastfeeding [5]. Findings from South Sudan showed that the prevalence of early initiation of breastfeeding ranged between 35% and 48% in both the hospital and community [6,7]. South Sudan is a young nation emerging from civil strife and has not implemented interventions such as the Baby-Friendly Hospital Initiative (BFHI) known to promote optimal breastfeeding [8,9,10].

The BFHI is a programme of the World Health Organization (WHO) and the United Nations Children’s Fund (UNICEF) which aims to protect, promote and support breastfeeding [11]. An analysis of systematic reviews on interventions to improve breastfeeding outcomes demonstrated that adhering to “the Ten Steps to Successful Breastfeeding” increased breastfeeding rates [8,12].

Given the low prevalence of early initiation of breastfeeding in our previous survey in Juba Teaching Hospital (JTH) [6], we carried out the BFHI training course for health workers to improve their competence to foster optimal breastfeeding. We hypothesized that the training would improve the knowledge, skills and attitude of staff and early breastfeeding practices of mothers. This survey assessed the effect of the BFHI training intervention on early initiation of breastfeeding at Juba Teaching Hospital in South Sudan.

## 2. Subjects and Methods

### 2.1. Design

This was a “before and after” study carried out among mothers at JTH in South Sudan. This study consisted of two parts, a baseline survey which forms the “before” part of this paper and the post-training intervention survey which forms the “after” part of this paper. The “before” survey was carried out in 2016 and has been published [6]. The rest of this section explains the details after the “before” part. The “after” survey was carried out between April–July 2018 after the training of health staff in December 2017.

#### Setting

This survey was carried out in the postnatal ward of JTH in South Sudan. This hospital conducts approximately 7000 deliveries annually. The average hospital stays are one day for normal births and three days for mothers who have had a caesarean section (CS).

### 2.2. Training of Health Workers in Juba Teaching Hospital

A paediatrician with specific training and practice in supporting lactation and breastfeeding conducted a 4-day training for 30 health workers including gynaecologists, doctors, midwives and nurses at the maternity unit of JTH in December 2017. All the health workers were enthusiastic and completed the 4-day training. The training was based on the UNICEF/WHO BFHI—a 20 hour course for maternity staff [13]. The training course consisted of 15.5 hours of theory and 4.5 hours of demonstrations, role-plays and hands-on practice.

Four to six months after the training, we carried out the “after” survey to assess breastfeeding practices in the same hospital. Between the “before” and “after” surveys, there was no change either in hospital leadership or in the maternity ward. No significant political or policy changes had taken place in the country in this field, nor had any interventions, such as breastfeeding awareness campaigns.

### 2.3. Inclusion and Exclusion Criteria

We included mothers who gave birth to live, healthy infants and excluded mothers who were discharged early, those who declined consent and those who had sick infants and infants with congenital malformations.

### 2.4. Sampling

A total of 990 mothers gave birth during both day and night to live infants in JTH during the survey. We excluded 116 mothers who were not contacted by the survey team because they had had an early discharge, 31 who declined consent and 33 who had sick infants and four infants with congenital malformations (Figure 1). We consecutively recruited 806 mother–infant pairs.

### 2.5. Variables

The dependent variable was early initiation of breastfeeding. Mothers were asked when they first put their babies on the breast after birth. Breastfeeding within one hour of birth was categorized as “early” and breastfeeding later than one hour after birth as “delayed”. The independent variables included the place of residence, marital status, mother’s age, education, employment, socioeconomic status, antenatal care, mode of birth, parity, infant sex, breastfeeding counselling during antenatal care, pre-lacteal feeding and discarding of colostrum.

### 2.6. Sample Size Estimation

We used the Open Epi calculator [14] and a formula for detecting differences between two proportions (Fleiss with continuity correction). The following considerations were made: an alpha of 5%, a power of 90%, a ratio of exposed to unexposed one and 48% prevalence of early initiation of breastfeeding from the baseline survey in JTH [6]. We assumed that the training would increase the prevalence of early initiation of breastfeeding to 56%. The calculated sample size was 803. We recruited 806 mother–infant pairs to make it comparable to the “before” survey.

### 2.7. Survey Instrument and Data Quality

We used the same questionnaire for the “after” survey as for the “before” survey. It was derived from the WHO guidelines [15] and an EBF survey in Eastern Uganda [16]. The study tool was initially pilot tested and amended accordingly. The principal investigator checked the collected data daily for completeness and consistency. The data were cleaned, coded, entered in an excel sheet and exported to STATA (Stata Corp, College Station, Texas, USA) version 15 for analysis.

### 2.8. Data Analysis

The two datasets, “before” and “after” survey datasets, were merged. We used conventional statistical methods to summarize and describe the data. We used a modified Poisson model to measure the effect of training on early initiation of breastfeeding and included, as confounders, variables that caused a variation of ≥10% between the crude and the adjusted prevalence ratio of effect of the training on early initiation of breastfeeding and known confounders from the literature. We used STATA version 15 (Stata Corp, College Station, Texas, USA) in the analysis of the data. 

### 2.9. Ethical Approval and Consent to Participate

We obtained ethical approval to conduct this research from the Directorate of Planning, Budgeting, and Research in the Ministry of Health in South Sudan (reference number: SMOH/E/JS/44. K.1) and the Norwegian Regional Committee for Medical and Health Research Ethics in the West (reference number: 2018/913/REK Vest). Official letters of permission were presented to the Director General of the hospital and ward in-charge. We obtained written informed consent from the mothers and a thumbprint from those who could not write. Privacy and confidential measures were maintained throughout the survey.

## 3. Results

### 3.1. Sociodemographic Characteristics 

A total of 806 mothers participated in the “after” survey. There were no differences in the sociodemographic characteristics of the participants before and after the training, as shown in Table 1. In the “after” survey, the mean age of the mothers was 24.8, with a standard deviation (SD) of 5.4. Less than half the mothers lived in rural areas, the majority were married, less than half had attended primary schools, most were not formally employed, and half had low socioeconomic status. Most mothers had attended antenatal care, less than half had had breastfeeding counselling during antenatal care, and majority had a normal and singleton birth.

### 3.2. Breastfeeding Practices 

As shown in Table 2, 48.1% (calculated by 388/806) of mothers practiced early initiation of breastfeeding before the training, compared to 90.8% (calculated by 732/806) of mothers after the training of health workers. The proportion of mothers discarding colostrum and/or giving pre-lacteal feeds after the training was significantly lower. The proportion of mothers who delivered babies by CS and adopted early initiation increased from 3% (calculated by 3/97) before the training of health workers to 60% (calculated by 12/20) after the training of health workers, as shown in Table 2. 

### 3.3. The Effect of Training on Breastfeeding 

In the bivariable analysis, the prevalence of early initiation of breastfeeding increased from 48.1% to 90.8% with a prevalence ratio of 1.89 (95% confidence interval CI: 1.75–2.03). In the multivariable analysis adjusting for marital status, mother’s residence, mother’s education, employment, socioeconomic status, antenatal care visits, mode of birth, parity and any breastfeeding counselling, the prevalence ratio was to some degree lower, which was 1.69 (95% CI: 1.57–1.82), as shown in Table 3. Regardless of the mode of birth, the training was effective on early initiation of breastfeeding at the bivariable and multivariable analysis.

## 4. Discussion

This survey assessed the effect of the BFHI training on early initiation of breastfeeding in JTH. In this paper, the prevalent of early initiation of breastfeeding increased from 48% before to 91% after health workers training. The mothers were 70% more likely to initiate breastfeeding early after health workers training. The increase in early initiation of breastfeeding could be due to the training of health workers since there were no other significant events in the country between 2016 and 2018 in this field. The training of the health workers might have mitigated barriers associated with delayed initiation of breastfeeding reported in previous study [6]. Further, this result showed that BFHI is still, even if it has been 25 years since its launch, an important programme to be implemented in places where it has not yet been implemented. Our findings are in accord with findings from a systematic review that revealed that health worker training programmes positively influence early initiation of breastfeeding [17]. Another study showed that training hospital nursery staff resulted in increased rates of early initiation of breast breastfeeding [18]. Regardless of the mode of birth, we found the BFHI training is effective in increasing the prevalence of early initiation of breastfeeding. This agrees with a report in Vietnam which showed that an intervention that improved health workers’ knowledge and skills increased the proportion of mothers who practiced early initiation of breastfeeding [19].

We found that more than half the mothers who gave birth by CS practiced early initiation of breastfeeding after the BFHI training. A study in Australia showed higher proportions of early initiation of breastfeeding among mothers who delivered babies by CS in BFHI hospitals [20]. However, targeted context-specific intervention is needed to further improve initiation of breastfeeding among mothers who give birth by CS in South Sudan [21].

In the current survey, the use of pre-lacteal feeds was lower after the training. This is possibly because of the training. This result is consistent with findings from a survey in India, which showed a decline in pre-lacteal feeding after breastfeeding education was offered by trained health staff [22]. Similarly, fewer mothers discarded colostrum after the training. This result is similar to findings from a survey in India, which showed an increase in mothers feeding colostrum to their infants after being educated and supported by trained health staff [22].

The strengths of this survey are as following: (1) it is the first to evaluate the effect of the BFHI training on early initiation of breastfeeding in South Sudan; (2) we used a large sample of mothers and carried out the survey 4–6 months after the training. However, the findings from this survey are difficult to generalize since the survey was carried out in one hospital. We did not ask the mothers’ views on early initiation and colostrum. We also did not assess health worker’s knowledge, attitude and skills after the training. The “before and after” survey design has its inherent internal validity limitations, such as historical records, reporting, testing and dropout threats. Furthermore, the mothers who were excluded from the study, including those who were not contacted due to early discharge, declined consent, sick infants and infants with congenital malformations, could have had different characteristics from those who participated in the survey.

## 5. Conclusions

Our findings suggest an urgent need to roll out the BFHI training to other hospitals in South Sudan. This will result in improved breastfeeding practices and ultimately improved maternal and infant health.

## Figures and Tables

**Figure 1 ijerph-16-03917-f001:**
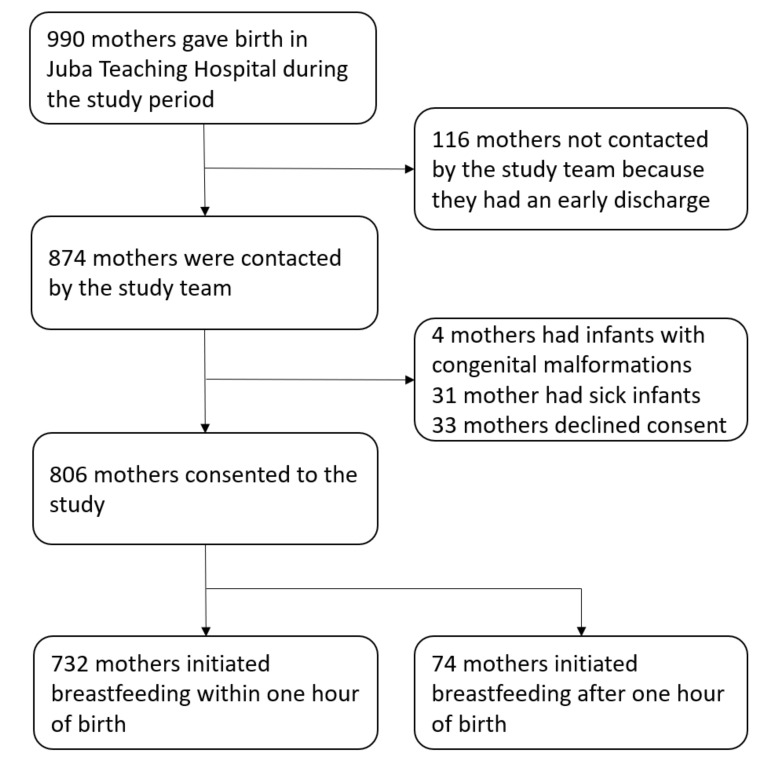
Profile of survey participants in April–July 2018 in Juba Teaching Hospital, South Sudan.

**Table 1 ijerph-16-03917-t001:** Sociodemographic characteristics of mother–infant pairs before and after the Baby-Friendly Hospital Initiative (BFHI) training in Juba Teaching Hospital, South Sudan.

Characteristics	Before BFHI training in 2016	After BFHI training in 2018
	N = 806	N = 806
	n (%)	n (%)
Place of residence		
Urban	410 (50.9)	351 (43.6)
Rural	396 (49.1)	455 (56.4)
Mother’s age		
15–19	139 (17.3)	153 (19.0)
20–24	252 (31.3)	256 (31.8)
25–29	246 (30.5)	227 (28.2)
30–34	123 (15.3)	120 (14.9)
≥35	46 (5.7)	50 (6.2)
Marital status		
Married	775 (96.2)	773 (95.9)
Single	31 (3.9)	33 (4.1)
Mother’s education		
None	132 (16.9)	166 (20.6)
Primary	377 (46.8)	322 (40.0)
Secondary	239 (29.7)	221 (27.4)
Tertiary	54 (6.7)	97 (12.0)
Mother’s employment		
Employed	179 (24.4)	139 (17.3)
Unemployed	609 (75.6)	667 (82.8)
Mother’s socioeconomic status		
Poorest	165 (20.5)	168 (20.8)
Poor	157 (19.5)	160 (19.9)
Medium	167 (20.8)	159 (19.3)
Richer	235 (29.2)	160 (19.9)
Richest	80 (10.0)	159 (19.7)
Infant sex		
Male	386 (47.9)	448 (55.6)
Female	420 (52.1)	358 (44.4)
Antenatal care		
0 visit	75 (9.3)	7 (0.9)
1–3 visits	311 (38.6)	326 (40.5)
≥4 visits	420 (52.1)	473 (58.7)
Mode of birth		
Normal	709 (88.0)	786 (97.5)
Caesarean section (CS)	97 (12.0)	20 (2.5)
Parity		
Single	790 (98.0)	793 (98.4)
Multiple	16 (2.0)	13 ( 1.6)
Breastfeeding counselling at antenatal care		
Yes	445 (55.2)	287 (35.6)
No	361 (44.8)	519 (64.4)

**Table 2 ijerph-16-03917-t002:** Breastfeeding practices among mother–infant pairs before and after the BFHI training in Juba Teaching Hospital, South Sudan.

Breastfeeding Practices	Before BFHI Training in 2016	After BFHI Trainingin 2018	*p*-Value
	N = 806	N = 806	
	n (%)	n (%)	
Initiation of breastfeeding			
Early initiation	388 (48.1)	732 (90.8)	<0.001
Delayed initiation	418 (51.9)	74 (9.2)	
Colostrum discarded			
No	739 (91.7)	782 (97.0)	<0.001
Yes	67 (8.3)	24 (3.0)	
Pre-lacteal feeding			
No	672 (83.4)	791 (98.1)	<0.001
Yes	134 (16.6)	15 (1.9)	
Caesarean section			
No	709 (88.0)	786 (97.5)	<0.001
Yes	97 (12.0)	20 (2.3)	

**Table 3 ijerph-16-03917-t003:** Bivariable and multivariable analysis among mother–infant pairs before and after the BFHI training in Juba Teaching Hospital in South Sudan.

Characteristics	Bivariable Prevalence Ratio (PR) (95% Confidence Interval, CI) N = 806	Multivariable PR (95% CI) N = 806
Training intervention		
Before	1	1
After	1.89 (1.75–2.03)	1.69 (1.57–1.82)
Place of residence		
Urban	1	
Rural	1.00 (0.94–1.07)	
Mother’s age		
15–19	1	
20–24	0.99 (0.90–1.08)	
25–29	0.92 (0.83–1.01)	
30–34	0.99 (0.89–1.10)	
≥35	0.94 (0.81–1.10)	
Marital status		
Married	1.32 (1.05–1.67)	1.28 (1.06–1.54)
Single	1	1
Mother’s education		
None	1	1
Primary	0.88 (0.81–0.96)	0.93 (0.87–1.00)
Secondary	0.89 (0.81–0.97)	0.92 (0.85–0.99)
Tertiary	0.99 (0.88–1.10)	0.94 (0.86–1.03)
Mother’s employment		
Employed	1	1
Unemployed	1.10 (1.00–1.20)	0.99 (0.92–1.07)
Child sex		
Male	1	
Female	1.00 (0.94–1.07)	-
Antenatal care		
0 visit	1	1
1–3 visits	2.10 (1.55–2.84)	1.48 (1.10–2.00)
≥4 visits	2.09 (1.54–2.83)	1.48 (1.10–1.99)
Mode of birth		
Normal	1	1
CS	0.17 (0.11–0.28)	0.22 (0.14–0.35)
Parity		
Single	1	1
Multiple	0.59 (0.38–0.91)	0.73 (0.48–1.11)
Breastfeeding counselling at antenatal care		
Yes	1	1
No	1.16 (1.08–1.24)	1.04 (0.98–1.10)

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
