# Peer review of "The Effect of Health Worker Training on Early Initiation of Breastfeeding in South Sudan: A Hospital-based before and after Study"

_ijerph, 2019, doi:10.3390/ijerph16203917_

Round 1

Reviewer 1 Report

This further paper of your research is interesting to read. Two overall points:

A. I am not following how staff training alone can result in this major change of practice. Was the lack of knowledge and attitude of the health workers the only barrier to these practices? (What general training did these health workers have that they were not aware of practice recommendations widely circulated for 25 years at least?) Previously, did mothers want to have early initiation and to give colostrum but health workers prevented this?
Were mothers asked their views on early initiation and colostrum? 
If your previous publications referred to reasons mothers gave for delayed initiation or not giving colostrum, then cite this/these previous publication(s) - don't assume readers know that data from this setting/study has been published previously.

B. Table 1: what does "breastfeeding counselling" involve? Did it change between the two time periods? The percentage receiving counselling dropped but the practices improved - discuss?
What info did pregnant women receive about early initiation and colostrum?

Minor points needing clarification in my view:

In abstract, phrase as "healthy infants" as mothers of some infants were excluded.

I would find it clearer if phrased with the before first then the after training - such as ".. compared to /increased from 48% ... to 91% ..."

Line 67: Describe "training course". Is it the actual UNICEF/WHO Breastfeeding Promotion and Support in a Baby Friendly Hospital: a course for maternity staff (2009) https://www.unicef.org/nutrition/index_24850.html or a locally developed training or what does "based on mean"? Did the training include an assessment of the health workers' competence?
What percentage of staff completed the training?

Line 68: Define/describe "breastfeeding expert" who provided the staff training. A woman who has breastfeed her own eight children? A health worker who has specific training in assisting lactation and breastfeeding? A health worker who is interested in the topic and reads about it?

Line 137: Sample characteristics: Seems a large SD for age. Would mean and range be clearer?

Discussion: Check relevant of references. # 15 Focus of study was education of mothers not staff training; #16 is broad community services not maternity unit;  #17  is only reporting the prevalence of early initiation of breastfeeding and use of colostrum, it does not mention staff training.

Check paragraph starting at Line 183 - seems repetitive. Consider saying "health worker training" to be clear it is not mother education you are referring to.

Line 194: "treats"?

Line 195: (different) or who had "sick" infants or who did not consent to be questioned (early discharge was not the only exclusion criteria)

Conclusion; Focus on outcome. For example adding "... towards improving practices and the heath of infants and their mothers" (not training for the purpose of training alone)

Best wishes with your future work. 

Author Response

Response to reviewer comments

We thank the reviewers for their detailed and professional suggestions and comments. We are grateful and know that they will make our manuscript professional and easier to read. We have revised the manuscript according to the comments. We have made point by point responses as follows:

Reviewer 1

Comment:

This further paper of your research is interesting to read

Response:

Thank you for this kind comment

Comment:

I am not following how staff training alone can result in this major change of practice. Was the lack of knowledge and attitude of the health workers the only barrier to these practices?

Response:

The training of health workers has the potential to result in such drastic change because health workers have better knowledge and skill to support breastfeeding mothers as stated in our hypothesis: the training of health workers would improve staff knowledge, skills, attitude and early breastfeeding practices in our study setting. Line 57-58; page 2.

Comment:

What general training did these health workers have that they were not aware of practice recommendations widely circulated for 25 years at least?

Response:

The nurses and midwives had basic medical training for three years. While, the doctors and gynaecologists had six years of medical training. Although they might have acquired knowledge during the medical training, assisting a mother towards successful breastfeeding is a hands-on skill which requires sensitive listening and supportive counselling. These skills might have been lost due to lack of refresher training since most the health workers were lived in Juba throughout the successive civil unrest in the country with no exposure to the rest of the world

Comment:

Previously, did mothers want to have early initiation and to give colostrum, but health workers prevented this?

Response:

Our previous paper did not capture information about health workers preventing mothers to initiate breastfeeding or give colostrum. However, an in-depth study on discarding of colostrum was one of the recommendations in the previous paper

Comment:

Were mothers asked their views on early initiation and colostrum? 

Response:

The current study did not ask mothers’ views on early initiation and colostrum, and this is a limitation in this paper

Comment:

If your previous publications referred to reasons mothers gave for delayed initiation or not giving colostrum, then cite this/these previous publication(s) - don't assume readers know that data from this setting/study has been published previously

Response:

In previous study, reasons or barriers for delayed initiation or not giving colostrum have been cited. Line 191-192; page 7.

Comment:

Table 1: what does "breastfeeding counselling" involve?

Response:

Breastfeeding counselling involves providing women basic information about the importance of early initiation of breastfeeding including nutrition, health benefit to both the mother and infant, bonding and the protective benefits of colostrum to the infants during antenatal care and after birth.

Comment:

Breastfeeding counselling. Did it change between the two time periods?

Response:

The number of mothers who received breastfeeding counselling during antenatal care reduced in the after study.

Comment:

The percentage receiving counselling dropped but the practices improved - discuss?

Response:

The number women who had received breastfeeding counselling during the antenatal care reduced, Table 1. Unfortunately, we did not assess for breastfeeding counselling at birth and we acknowledged this as limitation

Comment:

What info did pregnant women receive about early initiation and colostrum?

Response:

Our intervention was targeted to the health workers and not the mothers. We did not capture the information given to mothers by the health workers.

Minor points needing clarification in my view:

Comment:

In abstract, phrase as "healthy infants" as mothers of some infants were excluded.

Response:

In the abstract, we have added the inclusion and exclusion criteria and the sentence now reads as follows: We included mothers who gave birth to live, healthy infants and excluded those who were discharge early, declined consent, had sick infants and infants with congenital malformations. Line 19-21; page 1.

We also carried out necessary corrections in section 2.3. Line 9192; page 3.

Comment:

I would find it clearer if phrased with the before first then the after training - such as ". compared to /increased from 48% ... to 91% ..."

Response:

This has been rewritten accordingly: Line 22-32; page 1 and line 183-185; page 7.

Comment:

Line 67: Describe "training course". Is it the actual UNICEF/WHO Breastfeeding Promotion and Support in a Baby Friendly Hospital: a course for maternity staff (2009) https://www.unicef.org/nutrition/index_24850.html or a locally developed training or what does "based on mean"?

Response:

The training course have been described and the sentences is rewritten and reads as follows: A paediatrician with specific training and practice in supporting lactation and breastfeeding conducted a 4-day training for health workers including gynaecologists, doctors, midwives, and nurses at the maternity unit of JTH in December 2017.The training was based on the UNICEF/WHO Baby Friendly Hospital Initiative: a 20-hour course for maternity staff. The course consisted of 15.5 hours of theory and 4.5 hours of demonstrations, role-plays, and hands-on practice. Line 76-84; page 2.

Comment:

Did the training include an assessment of the health workers' competence?

Response:

Yes. The course included assessment of the workers competence

Comment:

What percentage of staff completed the training?

Response:

All the 30 selected staff completed the 4 days of training. Line 77-78; page 2.

Comment:

Line 68: Define/describe "breastfeeding expert" who provided the staff training. A woman who has breastfeed her own eight children? A health worker who has specific training in assisting lactation and breastfeeding? A health worker who is interested in the topic and reads about it?

Response:

We have defined "breastfeeding expert". Line 76; page 2.

Comment:

Line 137: Sample characteristics: Seems a large SD for age. Would mean and range be clearer?

Response:

We have corrected and now reads as follows: the mean age of the mothers was 24.8, with a standard deviation (SD) of 5.4. Line 151; page 4

Comment:

Discussion: Check relevant of references. # 15 Focus of study was education of mothers not staff training; #16 is broad community services not maternity unit; #17 is only reporting the prevalence of early initiation of breastfeeding and use of colostrum, it does not mention staff training.

Response:

We have checked references #15,16 and 17. We deleted irrelevant references and replaced them with appropriate one. Line 195-196; page 8.

Comment:

Check paragraph starting at Line 183 - seems repetitive. Consider saying "health worker training" to be clear it is not mother education you are referring to.

Response:

Done. Line 184-186; page 7.

Line 194: "treats"?

Response:

This ward "treats" has been deleted and replaced with “threats.” Line 221; page 8.

Comment:

Line 195: (different) or who had "sick" infants or who did not consent to be questioned (early discharge was not the only exclusion criteria).

Response:

We have rewritten the sentence and now reads as follows: Furthermore, possibly the mothers who were excluded from the study including those who were not contacted due to an early discharge, declined consent, mothers who had sick infants and infants with congenital malformations could have had different characteristics from those who participated in the survey. Line 222-224; page 8.

Comment:

Conclusion; Focus on outcome. For example, adding "... towards improving practices and the heath of infants and their mothers" (not training for the purpose of training alone)

Response:

Our findings suggest an urgent need to roll out the Baby-Friendly Hospital Initiative training to other hospitals in South Sudan. This will result in improved breastfeeding practices and ultimately improved maternal and infant health. Line 227-229; page 8.

Comment:

Best wishes with your future work

Response:

Thank you  

Reviewer 2 Report

The effect of health worker training on early initiation of breastfeeding in South Sudan: a hospital-based before and after study

Thank you for the opportunity to review this paper. It shows a significant improvement in the proportion of mothers who initiate breastfeeding within an hour of birth after hospital staff had received BFHI training.

Comments to the authors:

Page 2 line 67: It would be interesting to see some information on the attitudes of the staff to the BFHI training

Page 2 section 2.4: The previous publication from the group (the ‘before’ survey) emphasized the recruitment of mothers who delivered at all times of day and night. Was the same care taken with the current population?

Page 4: The last reference on page 3 is (13) and the first on page 4 is (19). References (14-18) are out of order in the manuscript.

Page 4 line 110: Please correct me if I am wrong, but the text refers to Eastern Uganda while reference (20) discusses Malawi.

Page 4 line 140: The statement that “at least half had had any breastfeeding counselling” seems at odds with the Table showing 35.6% in the ‘After’ group had Breastfeeding counselling.

Page 5 line 151: Data for early initiation among CS mothers is not presented in Table2.

Page 7 line 182: The authors claim the references (21) and (22) also found positive effects of BFHI training on early initiation of breastfeeding after CS. Reference (21) shows better rates in the BFHI hospital, but this is not necessarily cause and effect. Reference (22) only provides speculation that, “More targeted

interventions towards women planning on c-section delivery from the perinatal period and into the postpartum time, may encourage breastfeeding initiation and continuation of breastfeeding."

Page 7 line 183: Delete “high”

Page 8 line 194: Change “treats” to “rates”

Author Response

Response to reviewer comments

We thank the reviewers for their detailed and professional suggestions and comments. We are grateful and know that they will make our manuscript professional and easier to read. We have revised the manuscript according to the comments. We have made point by point responses as follows:

Reviewer 2

Comments and Suggestions for Authors

The effect of health worker training on early initiation of breastfeeding in South Sudan: a hospital-based before and after study

Thank you for the opportunity to review this paper. It shows a significant improvement in the proportion of mothers who initiate breastfeeding within an hour of birth after hospital staff had received BFHI training.

Response:

Thank you for your kind comments.

Find below are our point-by-point responses

Comment:

Page 2 line 67: It would be interesting to see some information on the attitudes of the staff to the BFHI training

Response:

We have not assessed health workers attitude. However, we observed that all the health workers were enthusiastic and completed the 4 days of training. Line 78-79; page 2.

Comment:

Page 2 section 2.4: The previous publication from the group (the ‘before’ survey) emphasized the recruitment of mothers who delivered at all times of day and night. Was the same care taken with the current population?

Response:

Yes, during the after survey, we contacted all the mothers who gave birth both at day and night during the study period. Line 96; page 3.

Comment:

Page 4: The last reference on page 3 is (13) and the first on page 4 is (19). References (14-18) are out of order in the manuscript.

Response:

Thank you for the observation. We have updated the references

Comment:

Page 4 line 110: Please correct me if I am wrong, but the text refers to Eastern Uganda while reference (20) discusses Malawi.

Response:

Thank you for the correct observation. We have included the correct reference and updated the references. Line 124; page 4.

Comment:

Page 4 line 140: The statement that “at least half had had any breastfeeding counselling” seems at odds with the Table showing 35.6% in the ‘After’ group had Breastfeeding counselling.

Response:

We have corrected that, and the sentence now reads as follows: less than half had had any breastfeeding counselling during antenatal care. Line 154; page 4.

Comment:

Page 5 line 151: Data for early initiation among CS mothers is not presented in Table2.

Response:

The data for early initiation among CS mothers has been included in Table 2.

Comment:

Page 7 line 182: The authors claim the references (21) and (22) also found positive effects of BFHI training on early initiation of breastfeeding after CS. Reference (21) shows better rates in the BFHI hospital, but this is not necessarily cause and effect. Reference (22) only provides speculation that, “More targeted interventions towards women planning on c-section delivery from the perinatal period and into the postpartum time, may encourage breastfeeding initiation and continuation of breastfeeding." 

Response:

We have rewritten the sentence and now reads as follows: We found that more than half of the mothers who gave birth by caesarean section practiced early initiation of breastfeeding after the BFHI training. A study in Australia showed better rates of early initiation of breastfeeding among mothers who delivered by CS in BFHI hospital. Line 205-206; page 8.

Comment:

Page 7 line 183: Delete “high”

Response:

The ward high has been deleted. Line 209; page 8

Comment:

Page 8 line 194: Change “treats” to “rates”

Response:

This ward "treats" has been deleted and replaced with “threats.” Line 221; page 8
